# Myeloma–Bone Interaction: A Vicious Cycle via TAK1–PIM2 Signaling

**DOI:** 10.3390/cancers13174441

**Published:** 2021-09-03

**Authors:** Takeshi Harada, Masahiro Hiasa, Jumpei Teramachi, Masahiro Abe

**Affiliations:** 1Department of Hematology, Endocrinology and Metabolism, Tokushima University Graduate School of Biomedical Sciences, Tokushima 770-8503, Japan; takeshi_harada@tokushima-u.ac.jp; 2Department of Orthodontics and Dentofacial Orthopedics, Tokushima University Graduate School of Biomedical Sciences, Tokushima 770-8504, Japan; mhiasa@tokushima-u.ac.jp; 3Department of Oral Function and Anatomy, Graduate School of Medicine Dentistry and Pharmaceutical Sciences, Okayama University, Okayama 700-8525, Japan; jumptera@okayama-u.ac.jp

**Keywords:** multiple myeloma, bone microenvironment, bone remodeling, TAK1, PIM2

## Abstract

**Simple Summary:**

Myeloma cells interact with their ambient cells in the bone, such as bone marrow stromal cells, osteoclasts, and osteocytes, resulting in enhancement of osteoclastogenesis and inhibition of osteoblastogenesis while enhancing their growth and drug resistance. The activation of the TAK1–PIM2 signaling axis appears to be vital for this mutual interaction, posing it as an important therapeutic target to suppress tumor expansion and ameliorate bone destruction in multiple myeloma.

**Abstract:**

Multiple myeloma (MM) has a propensity to develop preferentially in bone and form bone-destructive lesions. MM cells enhance osteoclastogenesis and bone resorption through activation of the RANKL–NF-κB signaling pathway while suppressing bone formation by inhibiting osteoblastogenesis from bone marrow stromal cells (BMSCs) by factors elaborated in the bone marrow and bone in MM, including the soluble Wnt inhibitors DKK-1 and sclerostin, activin A, and TGF-β, resulting in systemic bone destruction with loss of bone. Osteocytes have been drawn attention as multifunctional regulators in bone metabolism. MM cells induce apoptosis in osteocytes to trigger the production of factors, including RANKL, sclerostin, and DKK-1, to further exacerbate bone destruction. Bone lesions developed in MM, in turn, provide microenvironments suited for MM cell growth/survival, including niches to foster MM cells and their precursors. Thus, MM cells alter the microenvironments through bone destruction in the bone where they reside, which in turn potentiates tumor growth and survival, thereby generating a vicious loop between tumor progression and bone destruction. The serine/threonine kinases PIM2 and TAK1, an upstream mediator of PIM2, are overexpressed in bone marrow stromal cells and osteoclasts as well in MM cells in bone lesions. Upregulation of the TAK1–PIM2 pathway plays a critical role in tumor expansion and bone destruction, posing the TAK1–PIM2 pathway as a pivotal therapeutic target in MM.

## 1. Introduction

The skeletal system sustains a body to maintain physical functions. To maintain the vital roles of bone, bone homeostasis is strictly regulated by the balance between osteoclastic resorption and osteoblastic formation of bone. Osteoclasts (OCs) are monocyte-macrophage lineage cells of hematopoietic origin. OC precursors express receptor activator of nuclear factor-κB (RANK) induced by M-CSF. The binding of a RANK ligand (RANKL) to RANK activates the nuclear factor of activated T cells cytoplasmic 1 (NFATc1) to induce dendritic cell-specific transmembrane protein (DC-STAMP) and OC-stimulatory transmembrane protein (OC-STAMP) to form multinucleated cells while differentiating into mature OCs. RANKL is expressed on the surface of cells and also produced as a soluble form [1]. Osteoprotegerin (OPG), a soluble decoy receptor for RANKL, is produced by various cell types to suppress osteoclastogenesis [2]. Mesenchymal stem cells (MSCs)/bone marrow stromal cells (BMSCs) differentiate into osteoblasts (OBs) to form bone mainly through bone morphogenetic proteins (BMPs)-Smad and Wingless-type (Wnt)/β-catenin signaling pathways [3].

OCs and OBs communicate with each other in a basic multicellular unit (BMU) in the trabecular bone at bone remodeling phases [3,4]. The transition from bone resorption to bone formation is occurred by OC-derived coupling factors, which trigger osteoblastic differentiation and activation in resorbed lacunae to refill them with new bone. Most mature OBs eventually undergo apoptosis, while remaining OBs give rise to osteocytes in the bone matrix. Osteocytes are long-lived and the most abundant type in bone. Osteocytes embedded in the bone matrix are major sensors of mechanical stress to regulate bone remodeling through interaction with bone marrow cells by their dendritic processes [5,6]. Osteocytes regulate both OCs and OBs by elaborating molecules critical for bone metabolism, including RANKL and its inhibitor OPG, and the soluble inhibitors of Wnt/β-catenin signaling pathway sclerostin (SOST) and Dickkopf-1 (DKK-1) [7,8,9]. Osteocytes are generally accepted to be master regulators of normal and physiological bone remodeling. Thus, the direct and indirect crosstalk between osteocytes, OCs, and OBs in the BMU constantly regulates bone remodeling.

Bone also provides a unique microenvironment for not only normal hematopoietic cells but also malignant cells, including leukemic cells and multiple myeloma (MM) cells. The process of bone remodeling is skewed in various pathological conditions. Among others, MM cells uniquely expand preferentially in bone and form devastating bone destruction. In this review, we will discuss the underlying mechanisms for tumor expansion and bone destruction in MM and approaches to target the MM–bone interaction.

## 2. Bone Destruction in MM

MM cells stimulate bone resorption by enhancing osteoclastogenesis and suppress bone formation by inhibiting osteoblastic differentiation from BMSCs, leading to extensive bone destruction with rapid loss of bone (Figure 1) [10,11]. MM cells aberrantly over-produce bone resorptive cytokines, including macrophage inflammatory protein (MIP)-1α and MIP-1β, to upregulate RANKL in BMSCs, and the RANKL overexpression causes extensive osteoclastogenesis and bone resorption in MM [12,13,14]. In addition, osteoblastic differentiation is suppressed by factors overproduced from MM cells and/or their surrounding microenvironment in bone, including Wnt inhibitors (DKK1, sclerostin, and sFRP2,3), TNFα, IL-7, IL-3, TGF-β, and activin A [15,16,17]. These multiple factors cooperatively act and eventually develop bone-destructive lesions in MM.

## 3. MM Niche

Bone provides a unique microenvironment to foster MM cells as well as their stem cells. Preferential expansion of MM cells in the bone marrow indicates the critical role of the bone marrow microenvironment in MM cell growth and survival. MM cells acquire growth advantage by genetic alterations and clonal selection during disease progression. Furthermore, the cellular microenvironment in the bone lesions endows MM cells with additional growth potential and drug resistance. The cell types surrounding MM cells in bone lesions, including BMSCs, OCs, and vascular endothelial cells, create a beneficial cellular environment as a feeder for MM cell growth and survival. Such a cellular environment can be called the “MM niche” (Figure 1). MM cells impair a normal bone remodeling to cause bone destruction; crosstalk between MM cells and the microenvironment skewed in bone lesions forms a vicious cycle between MM tumor growth and bone destruction (Figure 1) [10,18]. “MM niche” also affects the function of immune cells. Although normal plasma cells marginally express programed cell death ligand 1 (PD-L1), its expression is elevated in MM cells to suppress immune function in effector T cells [19,20,21]. In addition, PD-L1 is expressed in not only MM cells but also cells surrounding MM cells in the bone marrow, including BMSCs, OCs, plasmacytoid dendritic cells (pDCs), and myeloid-derived suppressor cells (MDSCs) [20,22,23]. Thus, the immune function appears to be suppressed along with MM progression in the “MM niche”.

### 3.1. BMSCs/OBs

The adhesion of MM cells to BMSCs with defective osteoblastic differentiation via the very late antigen-4 (VLA-4) (α4β1 integrin)-vascular cell adhesion molecule-1 (VCAM-1) interaction confers MM cell homing, growth, survival, and cell adhesion-mediated drug resistance [24,25]. The adhesion stimulates MIP-1 production from MM cells, and MIP-1 acts on its cognate receptors, CCR1 and CCR5, expressed on MM cells in an autocrine and/or paracrine fashion to further activate VLA-4 and thereby enhance MM cell adhesion to BMSCs, forming a positive feedforward loop between the MM cell-BMSC adhesion and MIP-1 production by MM cells [26]. Furthermore, the MM cell adhesion stimulates BMSCs to produce various growth and anti-apoptotic factors for MM cells, such as IL-6, while inducing RANKL to enhance osteoclastogenesis. In contrast to the pro-MM factors over-produced in the bone marrow in MM, a murine model permissive for MM growth revealed reduction of host BMSC-derived adiponectin [27]. Adiponectin was found to induce MM cell apoptosis, and tumor burden and bone disease were enhanced in MM models using adiponectin-deficient mice. Interestingly, adiponectin was reported to be decreased in the serum of patients with monoclonal gammopathy of undetermined significance (MGUS) who subsequently progressed to MM [27]. Adiponectin is the most prevalent adipokine secreted by adipocytes; therefore, adipocytes may affect MM growth and survival in the bone marrow.

In addition, studies with xenograft and/or syngeneic MM mouse models revealed that quiescent MM cells reside in the endosteal osteoblastic regions more preferentially than in the perivascular regions in the bone marrow after intravenous inoculation of MM cells [28,29]. MM cells resided within the osteoblastic niche exhibited MM stem cell-like properties with in vitro colony formation and tumorigenic activity in secondary xenograft assays [28]. Endosteal OBs have been demonstrated to serve a niche to foster dormant MM cells [29]. These are reminiscent of the osteoblastic niches for normal hematopoietic as well as leukemic stem cells.

### 3.2. Osteocytes

Osteocytes are embedded in a mineralized bone matrix and represent the majority (90 to 95%) of all bone cells [30]. OBs synthesize osteoid matrix proteins, and then a portion of them are embedded in the osteoid matrix and differentiate into osteocytes. Osteocytes have multiple functions as mechanical sensors and endocrine cells. As sensors of mechanical loading, osteocytes regulate bone remodeling in physiological and pathological conditions via elaborating factors, including RANKL, OPG, sclerostin, and DKK-1 [7,31,32].

MM cells induce apoptosis in osteocytes through activation of Notch signaling in osteocytes [33]. Indeed, the number of viable osteocytes was decreased in bone specimens from patients with MM compared to those from normal subjects [34]. RANKL production is increased in osteocytes undergoing apoptosis in mice bearing MM cells, suggesting a critical role of osteocytes in the acceleration of osteoclastogenesis in MM [33]. The proteasome inhibitor bortezomib has been reported to maintain osteocyte viability by reducing both apoptosis and autophagy in osteocytes [35].

Sclerostin is a soluble inhibitor of the Wnt/β-catenin signaling pathway, and it blocks osteoblastic differentiation (Figure 1) [36,37]. Administration of anti-sclerostin monoclonal antibody, as well as genetic depletion of *Sost*, prevent MM-induced bone disease without apparent inhibition of MM tumor growth, using a xenograft model of MM [38]. Sclerostin is produced almost exclusively by osteocytes in normal subjects; however, circulating sclerostin is elevated in patients with symptomatic MM compared to MGUS [39]. Notably, MM cells were found to trigger the production of sclerostin even in OBs while inhibiting osteoblastic differentiation [40]. Since sclerostin can serve as a therapeutic target for myeloma bone disease, preclinical study reveals that anti-sclerostin monoclonal antibodies inhibit tumor growth and bone disease in combination with the proteasome inhibitor carfilzomib [40]. Regarding therapeutic strategies for the prevention of bone loss, the combination of an anti-sclerostin monoclonal antibody with zoledronic acid can be expected [41].

Osteocytes were recently reported to produce VEGF-A [42]. Its expression is enhanced by fibroblast growth factor (FGF23) from osteocytes under a hypoxic setting or cocultures with MM cells, suggesting promotion of angiogenesis and thereby MM progression by osteocytes. Furthermore, viable osteocytes have been demonstrated to decrease in number in MM [33]. Osteocytes protrude their dendrites into the bone marrow lumen and physically interact with MM cells to induce apoptosis in osteocytes in parallel with the production of RANKL and the soluble Wnt inhibitors sclerostin and DKK-1 to further exacerbate bone destruction. The role of osteocytes in MM-induced bone destruction remains to be further clarified.

### 3.3. OCs

OCs, highly specialized cells that resorb bone, reside in bone remodeling compartments, namely BMUs, which are covered by canopy cells in normal trabecular bones. However, the canopy cells disappear, and bone remodeling compartments are disrupted in MM, and MM cells go into the bone remodeling compartments and directly interact with activated OCs. When MM cells were cocultured with OCs, MM cell growth was enhanced [43,44]. MM cell proliferation was more potent in cocultures with OCs than those with BMSCs, while MM cells more firmly adhere to BMSCs and confer drug resistance [44]. Similar to BMSCs, OCs protect MM cells from apoptosis induced by anticancer drugs. The tumor necrosis factor (TNF) family members B-cell activating factor (BAFF) and a proliferation-inducing ligand (APRIL) have been demonstrated to be important survival factors for MM cells [45,46]. These factors are predominantly produced by OCs in the bone marrow in MM [47]. MM cells underwent apoptosis by TACI-Fc, a decoy receptor for both BAFF and APRIL in cocultures with OCs, suggesting MM cell growth and survival by OC-derived BAFF and APRIL [48,49].

Cancer cells generally overexpress hexokinase II (HKII), a key enzyme of glycolysis [50]. Most MM cells also overexpress HKII [51]. We reported that 3-bromopyruvate (3BrPA), a lactate analog to inhibit HKII, immediately suppresses ATP production to cause MM cell death [52]. Interestingly, enhanced Akt phosphorylation preferentially occurred in MM cells in cocultures with OCs but not BMSCs, which further increased HKII protein levels and lactate production by MM cells. The PI3K-Akt signaling has been demonstrated to protect HKII from its degradation to increase HKII protein levels in cells [53]. Consistently, the PI3K inhibitor LY294002 reduced HKII levels, and lactate production increased in MM cells in cocultures with OCs, suggesting activation of the PI3K-Akt-HKII pathway and thereby glycolysis in MM cells by OCs. Treatment with 3BrPA was able to efficiently kill MM cells in the presence of OCs. Furthermore, we observed that 3BrPA-induced ATP depletion abolishes ATP-binding cassette (ABC) transporter activity and retains drugs in MM cells even in cocultures with OCs, suggesting inactivation of ABC transporters in MM cells through blockade of glycolysis.

From these observations, OCs appear to significantly contribute to MM cell aggressiveness and drug resistance, providing a rationale for therapeutic strategies targeting the OC–MM interaction (Figure 1).

### 3.4. Vascular Endothelial Cells

Angiogenesis is progressively increased in the bone marrow in MM in parallel with bone destruction and tumor progression [46,54]. Not only MM cells but also BMSCs robustly secrete various angiogenic factors, including vascular endothelial growth factor (VEGF) and basic fibroblast growth factor [55,56].

Interestingly, OCs secrete a large amount of the proangiogenic factor osteopontin, a ligand of α_V_β_3_ integrin, and osteopontin from OCs and VEGF from MM cells cooperatively enhance angiogenesis [57]. Vascular endothelial cells express the VEGF receptor VEGF-R2 and α_V_β_3_ integrin; after the binding of their respective ligands, these receptors mutually interact with each other to potentiate their downstream signaling for angiogenesis. Osteopontin is known to be enzymatically cleaved and become functionally active [58]. OC-derived matrix metalloproteinase (MMP)-9 has been demonstrated to contribute to angiogenesis enhanced by OCs [59]. MMP-9 may affect the activity of osteopontin elaborated in bone lesions in MM. MM cells, OCs, and vascular endothelial cells can form a vicious cycle to aggravate bone destruction, angiogenesis, and MM expansion in MM bone lesions (Figure 1).

### 3.5. Adipocytes

MM cell–adipocyte interaction has also recently been highlighted in terms of MM progression. Mesenchymal stem cells (MSCs) give rise to both adipocytes and OBs. Their differentiation appears to shift toward adipogenesis but not to osteoblastogenesis upon interaction with MM cells [60]. Bone marrow adiposity is therefore increased at the early stage of MM, and adipocytes secrete multiple factors for MM cell evasion of apoptotic insult and promotion of MM cell migration [61]. The MM cell–adipocyte interaction induces a senescent-like phenotype in adipocytes [62]. Bone marrow adipocytes have been reported to be reprogramed and alter their phenotype by reduction of the nuclear receptor PPARγ through its promotor methylation by EZH2 and expression of its target genes [63]. The reprogramed adipocytes induce osteoclastogenesis and suppress osteoblastogenesis while producing different adipokines [63]. Furthermore, Li Z et al. have recently reported that adipocyte-secreted angiotensin II causes the upregulation of acetyl-CoA synthetase 2 (ACSS2) in MM cells, contributing to MM progression through stabilization of IRF4, an MM cell master regulator of gene expression [64]. Thus, BM adipocytes are thought to be deeply implicated in MM pathogenesis, and further studies are warranted.

### 3.6. MM Tumorigenicity and Plasticity in the Bone Marrow

Although cancer stem cells (CSCs) are accepted as a predominant cause of drug resistance in various types of cancers, the presence of CSCs are still conceptual in MM. A clonogenic capacity has been demonstrated to be possessed in phenotypically distinct MM plasma cell fractions and clonotypic B cells [65,66], leading to long-lasting controversies on CSCs or cancer-initiating cells in MM. MM CSCs give rise to mature MM cells. However, the clonogenic capacity of differentiated MM cells cannot be explained only with this one-way hierarchical model from MM CSCs to differentiated MM cells. MM cells and their CSCs are not static populations and transit between stem-like and non-stem-like states with phenotypically and functionally different cell types in the bone marrow and extraosseous local microenvironments. Human MM cells have been reported to expand exclusively in bone marrow cavities of subcutaneously implanted human or rabbit bones in SCID-hu or SCID-rab models, respectively, suggesting niches for MM CSCs in these experimental models [65,67].

Furthermore, Yaccoby et al. reported that in cocultures with OCs, mature MM cells with high CD138 expression gave rise to plasmablastic ones with reduced CD38 and CD138 expression and acquired drug resistance [68]. Chaidos et al. demonstrated conversion of mature MM cells isolated from patients to immature CD138^−^ pre-plasma cells and CD138^low^ plasma cells in SCID mice after mature CD138^+^ MM cells were injected via tail veins [69]. These immature clonotypic cells preferentially appeared in the spleen and liver, while mature MM cells composed the majority of clonotypic cells in the bone marrow. Circulating clonotypic B cells were also found to increase in number at diagnosis or at relapse but also persist in causing MM tumor recurrence even in patients with MM achieving a molecular remission in bone marrow samples after treatments [69]. Therefore, circulating clonotypic B cells may be a reservoir for MM-initiating cells to cause relapse. These observations collectively suggest an inverse hierarchical model from differentiated MM cells to immature MM clonotypic cells or MM progenitors in certain extra-osseous microenvironments rather than in the bone marrow. Further studies are needed to explore plasticity between phenotypically distinct MM plasma cell and clonotypic B cell fractions upon interaction with their preferential microenvironment.

### 3.7. The Role of Exosomes and miRNA

Exosomes are small extracellular vesicles with a membrane structure that express characteristic surface markers, proteins, and lipids on their surfaces and carry bioactive molecules, including miRNAs, growth factors, cytokines, and signaling molecules to target cells. Exosomes are released by various cells and play an important role in cell–cell communication between MM cells and the bone marrow microenvironment, thereby contributing to the pathogenesis and progression of MM. MM cell-derived exosomes have been demonstrated to be enriched in critical factors to suppress osteoblastogenesis, including lncRNA RUNX2-AS1 and DKK-1, and those to enhance osteoclastogenesis, including amphiregulin [70,71]. BMSC-derived exosomes in patients with MM can, in turn, promote the proliferation of MM cells [72]. MM cell-derived exosomes can also affect immune cells in the bone marrow microenvironment in MM. One such example is the induction of the immunosuppressive phenotype of MDSCs to suppress the effector function of T cells [73]. These findings collectively indicate the importance of exosome-mediated communication in MM cell–bone marrow interaction to enhance tumor progression/drug resistance and bone destruction in MM.

miRNAs are short non-coding RNAs that play a critical role in various cellular processes. miRNAs are aberrantly expressed or functionally deregulated in MM cells and microenvironmental cells in the bone marrow. Besides their upregulation in cells, miRNAs are abundantly contained in exosomes and carried from one cell to another. Numerous miRNAs have been demonstrated to be involved in the interaction between MM cells and the bone marrow microenvironment. Among these miRNAs, miRNAs such as miR-21-5p increase the RANKL/OPG ratio in bone marrow stromal cells in patients with MM [74], while others, including miR-342-3p, miR-363-5p, and miR-203a-3p, suppress osteoblastogenesis through targeting the RUNX2, BMP/SMAD, and canonical WNT/β-catenin pathways [75,76]. The aberrant miRNA expression and function have recently received a great deal of attention in the context of mutual interaction between MM cells and the bone marrow microenvironment [77].

### 3.8. The Effects of MM Bone Microenvironment on Immune Homeostasis

MM tumor microenvironments are created with pathologically skewed tumor-associated cells, which results in attenuation of immune cell function. BMSCs or MSCs are a predominant cell type to support MM cell growth and survival in the bone marrow. In addition, these cells potently suppress the activity and function of cytotoxic effector T cell and natural killer (NK) cell populations while inducing myeloid-derived suppressor cells (MDSCs), tumor-associated macrophages, CD38-positive regulatory B-cells (Bregs), and T cells (Tregs).

γδT cells are important effectors in the first-line defense against infections and tumors. Vγ_9_Vδ_2_ γδT cells represent a majority of γδT cells in peripheral blood and bone marrow in humans. We reported that recombinant non-peptide phosphoantigens, as well as zoledronic acid, are able to expand and activate γδT cells from human peripheral blood mononuclear cells (PBMCs) ex vivo in the presence of IL-2 or immunomodulatory drugs, namely lenalidomide and pomalidomide [78]. The surface expression of the cytotoxicity-associated molecules NKG2D, DNAX accessory molecule-1 (DNAM-1; CD226), and TRAIL as well as LFA-1, and intracellular granzyme B and perforin levels are upregulated in the expanded γδT cells, indicating the cytotoxic Th1-like nature of these cells. The expanded Th1-like Vγ_9_Vδ_2_ γδT cells exert potent cytotoxic activity against MM cell lines as well as primary MM cells ex vivo. However, their in vitro cytotoxic activity against MM cells is reduced in the presence of BMSCs even in high E:T ratios. Interferon-γ production and the surface expression of the cytotoxic molecules, NKG2D and DNAM-1, in the expanded Th1-like Vγ_9_Vδ_2_ γδT cells, are suppressed under cocultures with BMSCs. Castella et al. nicely reviewed the bone marrow microenvironments blunt γδT cell activity and function from the clinical stages of MGUS to MM and MM relapse and progression, although γδT cells reside in the bone marrow by their nature [79].

Besides BMSCs, a number of reports have demonstrated the immunosuppressive aspects of OCs. Mature OCs express various immune checkpoint molecules on their surface, including PD-L1, galectin-9, herpesvirus entry mediator, CD200, T-cell metabolism regulators indoleamine 2, 3-dioxygenase (IDO), and CD38, and can directly inhibit proliferation and activation of CD4-positive and CD8-positive effector T cells, while inducing immune regulatory cells, including FoxP3-positive regulatory T cells and MDSCs [20].

OCs as well as BMSCs also induce the expression of PD-L1 on MM cells, which further enhances the activation of programed cell death protein 1 (PD-1)/PD-L1 pathway to impair T cell proliferation and cytotoxicity against MM cells. Therefore, BMSCs and OCs induce and maintain an immunosuppressive milieu in bone marrow in patients with MM.

## 4. Targeting MM Cell–MM Niche Interaction

MM-induced cell types in MM bone lesions, namely OCs, vascular endothelial cells, and BMSCs with defective osteoblastic differentiation, create the “MM niche” to confer drug resistance to MM cells. To disrupt MM cell–MM niche interaction, the underlying mechanisms for tumor expansion and bone destruction in MM have been studied, and various factors become possible therapeutic targets against drug resistance elicited through the MM–bone interaction (Table 1).

### 4.1. Targeting of Osteoclastogenesis and Acidic Microenvironments

To prevent MM bone destruction, MM tumor reduction is the most important. In addition, inhibition of OC activity by bisphosphonates, RANK-Fc, and osteoprotegerin have been observed to prevent not only MM-induced bone destruction but also MM tumor expansion in MM animal models [18,80,81], suggesting that potent anti-resorptive therapy blunts a vicious cycle between osteoclastic bone destruction and MM tumor expansion.

Zoledronic acid has been demonstrated to prevent the occurrence of skeletal-related events (SRE) more efficaciously, improve median overall survival, and prevent SRE compared to clodronate, an oral less potent bisphosphonate [82]. These observations are consistent with the notion that interaction between OCs and MM cells may play an important role in MM expansion in the bone marrow. Denosumab is a fully human monoclonal antibody of the IgG_2_ subclass targeting RANKL with high affinity to prevent its interaction with RANK on OCs. A large phase 3 study has demonstrated that denosumab is non-inferior to zoledronic acid in newly diagnosed MM patients with at least one bone lesion(s) in terms of delaying time to first SRE and overall survival [83]. Thus, it is recommended to administer denosumab or zoledronic acid at the start of treatment in newly diagnosed MM patients with bone lesion(s). Denosumab is more strongly recommended in patients with renal impairment due to its low renal toxicity.

Under low O_2_ conditions and as a consequence of glycolysis (the Warburg effect), cancer cells highly produce protons and lactate, leading to extracellular acidification to pH 6.4–7.0, while pH values are 7.2–7.4 in normal tissues [84]. Activated OCs on the bone surface abundantly secrete protons into excavated pits (~pH 4–5) to resorb bone while acidifying their close vicinity [85]. In osteolytic bone lesions in MM, therefore, the MM cell-OC interaction appears to create a highly acidic milieu by protons produced by OCs and lactate by proliferating glycolytic MM cells. We reported that acid activates the PI3K-Akt signaling to upregulate the acid sensor TRPV1 in MM cells, thereby forming a positive feedback loop between acid sensing and the PI3K-Akt survival signaling [86]. In addition, tumor acidity has been demonstrated to blunt cytotoxic effects of various chemotherapeutic agents as well as the activity of immune effecter cells [87,88]. Therefore, acidic conditions should be targeted to improve the therapeutic efficacy against MM.

Reveromycin A (RM-A) is a small microbial metabolite with three carboxylic groups isolated from *streptomyces* sp. SN-593 [89,90]. In an acidic microenvironment, RM-A becomes a non-polar form, which is able to permeate a cell membrane and induce apoptosis by inhibiting isoleucine tRNA synthesis [89,90]. As such, RM-A has been demonstrated to preferentially induce apoptosis in acid-producing OCs but not in other types of normal cells [89,90,91]. We recently reported that RM-A induced caspase-dependent apoptosis in MM cells at acidic culture conditions but not at pH 7.4 and decreased the progression of a tumor as well as osteolytic lesions in bones in human SCID-rab MM animal models [92]. Because extracellular acidification makes RM-A permeate cell membrane to induce apoptosis, it is plausible that an acidic milieu created by the OC–MM cell interaction rather induce cytotoxic activity by RM-A against MM cells as well as acid-producing OCs. Of note, RM-A and bortezomib, in combination, almost completely eradicated MM cells and resumed bone formation in the animal MM models [92]. Although acidic conditions are thought to confer drug resistance, RM-A is suggested to target not only OCs but also MM cells in a highly acidic milieu in MM bone lesions, and its anti-MM effects can be envisaged to be augmented in combination with bortezomib which is active at non-acidic sites.

**Table 1 cancers-13-04441-t001:** Therapeutic targets for MM–bone interaction.

	Target Molecules	Agents under Preclinical Investigation	Clinically Available Agents
MM–OCs interaction	Mevalonate pathway RANK–RANKL CXCR4/SDF-1 MIP-1α IL-3 IL-17 BAFF PIM2, TAK1	CXCR4 inhibitor [93] CCR1 inhibitor [94] Anti-IL-3 mAb [95] Anti-IL-17A mAb [96] Anti-BAFF mAb [97] PIM inhibitor [98,99], TAK1 inhibitor [100]	Bisphosphonate [82] Anti-RANKL mAb [83]
MM–BMSC/OB interaction	Sclerostin DKK-1 Notch pathway IL-7 TGF-β Activin A Adiponectin PIM2, TAK1 RUNX2, ATF4	Anti-DKK1 mAb [101] γ-secretase inhibitor [102], HDAC inhibitor [103] Anti-IL-7 polyclonal Ab [104] TGF-β receptor inhibitor [17] Recombinant fusion protein of activin receptor type IIA and human IgG_1_ Fc domain [105] Apolipoprotein A1 mimetic peptide [27] PIM inhibitor [99,106], TAK1 inhibitor [100]	Anti-sclerostin mAb [107] (available for osteoporosis) Proteasome inhibitors

OCs—osteoclasts; OB—osteoblast; RANK—receptor activator of nuclear factor-κB; RANKL—RANK ligand; CXCR4—C-X-C motif chemokine receptor 4; SDF-1—stromal cell-derived factor 1; MIP-1α—macrophage inflammatory protein-1α; CCR1—C-C motif chemokine receptor 1; IL—interleukin; BAFF—B-cell activating factor; PIM2—proviral integration of Moloney leukemia virus-2; TAK1—TGF-β-activated kinase 1; TGF-β—transforming growth factor-β; DKK1—Dickkopf-1; RUNX2—runt-related transcription factor 2; ATF4—activating transcription factor 4.

### 4.2. The Role of MM Metabolism in the Bone Marrow

Glutamine (Gln) supports bioenergetics and biosynthesis and produces antioxidants through glutaminolysis in cancers, which makes them highly glutamine dependent for their survival and proliferation [108]. Gln levels have been demonstrated to be decreased in association with increased glutamate levels in bone marrow plasma in patients with active MM, indicating enhanced glutaminolysis [109]. Previous works nicely demonstrated the critical roles of MM-dependent alterations of Gln metabolism in the development of MM bone disease. MM cells appear to be deficient in glutamine synthetase and highly consume ambient Gln to reduce Gln levels in the bone marrow [110]. Gln also contributes to the energy production necessary for the active biosynthesis of the osteogenic matrix in OB precursors, and Gln deprivation suppresses in vitro OB differentiation of MSCs in osteogenic media. Importantly, OB differentiation is impaired even with a slight reduction of Gln content in cultures from 0.6 mM, the physiological level, to 0.4 mM, the average level of MM BM plasma, suggesting a critical involvement of such skewed Gln metabolism in patients with active MM in the progression of bone destruction. Gln provides carbon moieties for the synthesis of many other non-essential amino acids, including asparagine (Asn). Primary MSCs require Gln or Asn for their osteoblastogenesis; however, Asn but no other Gln-related non-essential amino acids can replace the role of Gln in osteoblastogenesis. Therefore, mitigation of Gln metabolism in MM cells and supplementation of Asn may become a novel approach to ameliorate bone destruction in patients with MM.

In addition to HKII, as described in Section 3.3, the pyruvate kinase (PK) isoform PKM2, a critical glycolytic enzyme in cancer cells, is also overexpressed in MM cells and play an important role in MM cell growth and proliferation [111]. The peroxisome proliferator-activated receptor γ coactivator 1α (PGC1α) and PGC1β play an important role in glucose, lipid, and energy metabolism [112]. Beharry et al. demonstrated the critical role of PIM kinases in the induction of PGC-1α-mediated mitochondrial biogenesis and energy production, thereby negatively regulating AMPK activation in lung cancer and leukemia cell lines [113]. Importantly, Zhang et al. nicely reported that PGC1β is highly expressed in different MM cells and that upregulation of PGC1β enhances glycolysis in MM cells and thereby MM cell proliferation [114]. These results demonstrate the importance of glycolysis and mitochondrial biogenesis and thereby energy production in MM cells. Although there have been a number of studies of metabolisms skewed in MM cells, further studies on metabolic features of MM cells in the bone marrow microenvironment will be emphasized in terms of drug resistance against new anti-MM agents.

### 4.3. Induction of Bone Formation

Proteasome inhibitors are able to exert bone anabolic actions in responders. Suppression of MM tumor growth has been reported to be inversely correlated with the elevation of serum levels of bone-specific alkaline phosphatase as a marker of osteoblastic activity in patients with MM treated with the proteasome inhibitors bortezomib and carfilzomib [115,116,117]. Such inverse correlation between bone anabolic response and tumor regression was recapitulated in MM animal models [118]. The emergence of bone anabolic activity with anti-DKK1 antibody [119,120], lithium chloride [121], and activin A inhibitor [122,123], as well as with the enforced expression of Wnt3a within bone [124], has also been observed in MM animal models. These observations suggest that osteoblastic differentiation and activity negatively impact MM tumor growth and that MM cells suppress osteoblastic differentiation to protect themselves from such OB-mediated growth suppression (Figure 1). Furthermore, bortezomib has been reported to maintain osteocyte viability and improve bone integrity in bone samples from MM patients [35]. A higher reduction of autophagic osteocytes was observed in MM patients treated with bortezomib-based regimens compared with those treated without bortezomib. Bortezomib was also found to increase the level of LC3II/I ratios and block the degradation of p62 in osteocyte-like cells in cocultures with MM cells as well as upon treatment with high-dose dexamethasone, suggesting suppression of the autophagic death of osteocytes by bortezomib. Therefore, proteasome inhibitors appear to maintain osteocyte viability and thereby improve bone integrity in MM patients.

TGF-β has been demonstrated to inhibit the terminal differentiation of OBs and their mineralization [125]. TGF-β is stored in bone matrices as its latent form, released from bone matrices through osteoclastic bone resorption and activated by acids and metalloproteinases secreted from OCs [126]. TGF-β appears to be abundantly released and activated in osteoclastic bone destructive lesions in MM.

We reported that inhibition of the TGF-β signaling resumes osteoblastic differentiation from BMSCs suppressed by MM cells and that thus induced OBs impair MM cell growth and survival and make MM cells susceptible to anti-MM agents [17]. These results are consistent with the notion that the impairment of osteoblastic differentiation or bone formation by MM cells not only causes rapid loss of bone together with enhanced bone resorption in MM but also provides favorable conditions with the undifferentiated BMSCs for MM growth and survival (Figure 1). Thus, TGF-β inhibition may become a novel approach to restore bone formation with tumor containment in MM.

Activin A, a TGF-β superfamily member, plays an important role in a variety of fundamental biological processes, including organ development, erythropoiesis, and hormone signaling. It is also involved in bone remodeling in the skeleton [127]. Activin A levels were reported to be increased in bone marrow plasma in MM patients with bone lesions [122]. MM cells activate the JNK pathway in BMSCs to enhance activin A production by BMSCs (Figure 1) [122]. Activin A activates Smad2 and thereby suppresses distal-less homeobox (DLX)–5 expression to inhibits osteoblastic differentiation from BMSCs.

### 4.4. Targeting PIM Kinases

A number of intracellular signaling mediators are activated in both MM cells and cells surrounding MM cells in the bone marrow, leading to aggravation of pathological bone loss and MM tumor expansion and dissemination. We extensively looked for factors responsible for MM tumor growth and survival in bone lesions and found the serine/threonine kinase PIM2 as a critical mediator in MM cell growth and survival [128]. PIM2 is constitutively overexpressed in MM cells and further upregulated in cocultures with OCs as well as BMSCs [128] (Figure 2). PIM2 was identified in the 34 most highly overexpressed genes in MM cell lines compared to lymphoma and AML cell lines [129]. Although PIM2 appears to be the most upregulated isoform of PIM family members in MM cells, MM cells also expressed PIM1 at various levels, but rarely PIM3. Transgenic mice overexpressing PIM1 or PIM2 were reported to develop lymphoma; therefore, these kinases have been regarded as oncogenic factors [130,131]. As normal quiescent bystander cells marginally express PIM2, PIM2 can serve as an MM-specific therapeutic target.

Besides PIM-2, PIM1 is also aberrantly expressed in a portion of MM cells, which should be targeted. Although tumor-suppressor miR-33b represses PIM1 expression, miR-33b expression has been demonstrated to be suppressed in primary CD138 positive plasma cells derived from MM patients [132]. PIM1 is also a direct target gene of miR-33a-5p. Wu L et al. have recently reported that long non-coding RNA LINC01003 functions as a sponge of miR-33a-5p to inhibit the development of MM by downregulating PIM1 expression [133]. They observed that MM cells express LINC01003 at low levels, which may allow their progression with enhanced PIM1 expression. Furthermore, PIM kinases are also involved in the expression and activation of drug efflux ABC transporters to confer drug resistance [134]. PIM1 phosphorylates the ABC transporter ABCG2/breast cancer resistance protein (BCRP) and thereby promotes its multimerization with its stable membrane expression [135]. Because ABCG2/BCRP is often overexpressed in MM cells, the overexpression of PIM kinases may be associated with increased drug efflux in MM cells to elicit drug resistance.

The IGF-1/PI3K/Akt pathway is regarded as an important therapeutic target in MM [136,137]. However, because the survival pathways mediated by PIM kinases seem independent of the PI3K/Akt pathway [128,138], PIM kinases and the PI3K/Akt pathway should be inhibited together to improve anti-MM efficacy.

Lu et al. reported that PIM2 phosphorylates TSC2, a negative regulator of mammalian target of rapamycin C1 (mTORC1), on Ser-1798 and relieves the suppression of TSC2 for mTORC1, thereby promoting MM cell proliferation [139]. 3-(S)-amino-piperidine pyridyl carboxamide (LGB321), a selective small-molecule pan-PIM kinase inhibitor, significantly decreased the mTORC1 activity to inhibit MM cell proliferation [139]. Subsequently, the pan-PIM kinase inhibitor PIM447 has been demonstrated to directly induce MM cell apoptosis with cell-cycle disruption, alongside of a bone-protective effect [99]. Oral administration of PIM447 has been demonstrated to be well tolerated and exert single-agent antitumor activity in relapsed/refractory MM patients [140]. In search for potential oral drug combination with PIM447, Paíno et al. recently reported that anti-MM efficacy can be improved by PIM447 in combination with pomalidomide plus dexamethasone (Pd) in preclinical studies [141]. This three-drug combination significantly improves survival than Pd alone [141]. IRF4, a master transcription factor for MM cell growth and survival, is regarded as a major target of pomalidomide; the PIM447-Pd treatment further downregulated IRF4 levels in MM cells. They also demonstrated that the combination PIM447-Pd potently inhibits mammalian target of rapamycin complex 1 and thereby eIF4E to potently suppress protein translation in MM cell. Clinical activity of combinatory treatment with PIM447 should be further studied. Efficacy of INCB053914, a novel pan-PIM kinase inhibitor, has been studied for various hematological malignancies [142]. The preclinical study revealed that INCB053914 additively or synergistically induces tumor cell death in combination with PI3Kδ inhibition, JAK1/2 inhibition, or cytarabine, proceeding to its clinical trials. PIM kinase inhibition may become a novel therapeutic approach in MM.

Interestingly, PIM2 expression is also upregulated as a negative regulator for osteoblastogenesis in BMSCs and preosteoblastic cells in the presence of MM cells, as well as cytokines, known as inhibitors for osteoblastic differentiation in MM, including TNF-α, IL-3, IL-7, TGF-β, and activin A (Figure 2). *PIM2* knockdown, as well as the PIM inhibitor SMI-16a, can restore osteoblastogenesis suppressed by MM cells and in the presence of these inhibitory factors [106]. We found that mature OCs on the surface of bone also express PIM2 in addition to MM cells in mouse MM models with intra-tibial inoculation of 5TGM1 MM cells [98]. RANKL is upregulated to enhance osteoclastogenesis in MM bone lesions. Treatment with RANKL was found to induce PIM2 expression in parallel with c-fos and NFATc1 in OC precursors; treatment with *PIM2* siRNA or the PIM inhibitor SMI-16a suppressed c-fos, NFATc1, and cathepsin K expression along with abolishing osteoclastogenesis by RANKL [98]. These results suggest the therapeutic efficacy of PIM inhibition for bone destruction in MM.

### 4.5. Targeting TAK1

TGF-β-activated kinase 1 (TAK1) is a member of the mitogen-activated protein kinase kinase kinase (MAP3K) family, also known as MAP3K7 [143,144]. It was originally identified as a key kinase in transducing TGF-β signaling down to p38 MAPK and c-Jun and N-terminal kinase (JNK) [143]. Subsequently, TAK1 has been demonstrated to be associated with the activation of a wide range of intracellular signaling pathways important for various cellular functions, including the activation of NF-κB and extracellular signal-regulated kinase (ERK) [143]. TAK1 appears to be a gatekeeper to facilitate the multiple important intracellular signaling pathways (Figure 3). Activation for TAK1 requires TAK1-binding proteins (TABs), and TAK1 activity is regulated by the complex of TAK1 and TAB1 or TAB2/3 [145]. TAB1 binds to a TAK1 kinase domain, while TAB2 or TAB3 binds to a region near the TAK1 c-terminus. Cellular stimuli lead to form the structure of lysine-63-linked poly-ubiquitinylated (pUb) chains, and TAB2/3 recognizes the pUb-chains, resulting in the catalytic activation of TAK1, followed by its auto-phosphorylation and full enzymatic activation. We recently reported that TAK1 is constitutively overexpressed and phosphorylated in MM cells and that TAK1 acts as an upstream regulator responsible for multiple signaling pathways critical for MM growth and survival, including PIM2-mediated ones (Figure 3) [100]. Of note, TAK1 phosphorylation is also induced in BMSCs in cocultures with MM cells, which facilitates MM cell adhesion to BMSCs via interaction between VLA-4 and VCAM-1, thereby inducing IL-6 production and RANKL expression by BMSCs [100]. In OC precursors, RNAKL upregulates cellular FLICE inhibitory protein (c-FLIP), which blocks death signaling from stimulation of tumor necrosis factor-related apoptosis-inducing ligand (TRAIL) [146]. Therefore, TRAIL does not induce apoptosis in OC-lineage cells but induces phosphorylation of TAK1 through a death receptor-mediated formation of complex II and activates NF-κB signaling pathway, resulting in enhancement of OC formation, survival, and activation. Importantly, TAK1 inhibition was able to effectively impair MM cell adhesion to BMSCs and induce MM cell death and abolished PIM2 expression induced in osteoclastic lineage cells and BMSCs to suppress osteoclastogenesis and restore the differentiation of BMSCs into bone-forming mature OBs, respectively. In addition, fibroblast growth factor receptor 3 (FGFR3) tyrosine kinase interacts with and activates TAK1 in MM cells, suggesting susceptibility of MM cells with t(4;14) to TAK1 inhibition [147]. Therefore, TAK1 appears to be a pivotal therapeutic target in MM to disrupt the key signal transduction pathways responsible for tumor progression and bone destruction.

## 5. Perspectives and Conclusions

The recent development of immunotherapies with therapeutic antibodies has been revolutionizing a treatment paradigm in MM. However, the efficacy of the immunotherapies is limited by the dysfunction of effector cells and the immunosuppressive microenvironment created by cells surrounding MM cells, including OCs and BMSCs with defective osteoblastic differentiation. Up to now, MM still remains difficult to be cured even after effectively debulking MM tumor with new treatment modalities, and MM bone disease also remains a significant clinical problem for which there is, as yet, no effective cure, albeit with attaining longer survival, keeping physical function is considered as a more important issue after repeated treatment, especially in elderly frail patients. In this regard, the development of new agents to regain bone mass and strengthen muscular function is wanted. Because TAK1 and PIM2-mediated signaling are vital for tumor expansion and bone destruction in MM, the development of novel agents to target TAK1 and/or PIM2 may play a role.

As to relapse even after deep response with recent treatment modalities, there is the idea that MM CSCs have an innate resistance to chemotherapeutic agents and regrow in local microenvironments in and outside of the bone marrow. We need to clarify where MM CSCs localize and how they self-renew upon interaction with the MM microenvironment. Further elucidation of the molecular mechanisms of MM cell and/or MM CSC–bone interactions will provide us with new approaches that have a real impact on both bone disease and tumor progression in MM.

## Figures and Tables

**Figure 1 cancers-13-04441-f001:**
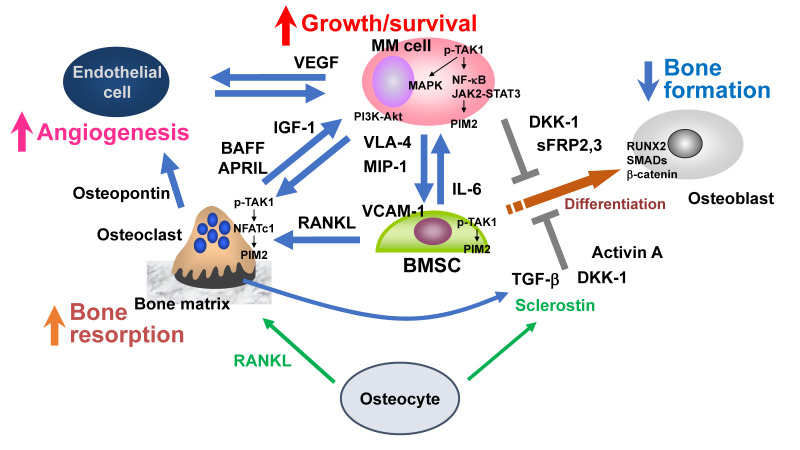
Bone destruction and MM niche. Myeloma (MM) cells stimulate bone resorption by enhancing osteoclastogenesis while suppressing bone formation by inhibiting osteoblastic differentiation from bone marrow stromal cells (BMSCs), leading to extensive bone destruction with rapid loss of bone. The TAK1–PIM2 axis is among major crucial signaling pathways in MM cells, BMSCs, and osteoclasts for MM progression and MM-related bone loss. Osteocytes embedded in the bone matrix produce RANKL and sclerostin to regulate bone metabolism. RANKL and sclerostin production by osteocytes are aberrantly increased in MM. Angiogenesis is also increased by multiple factors elaborated from osteoclasts and BMSCs as well as MM cells. MM-induced cell types in MM bone lesions, namely osteoclasts, vascular endothelial cells, and BMSCs with defective osteoblastic differentiation, create cellular microenvironments suitable for MM growth and survival and confer drug resistance to MM cells, which can be called an “MM niche”.

**Figure 2 cancers-13-04441-f002:**
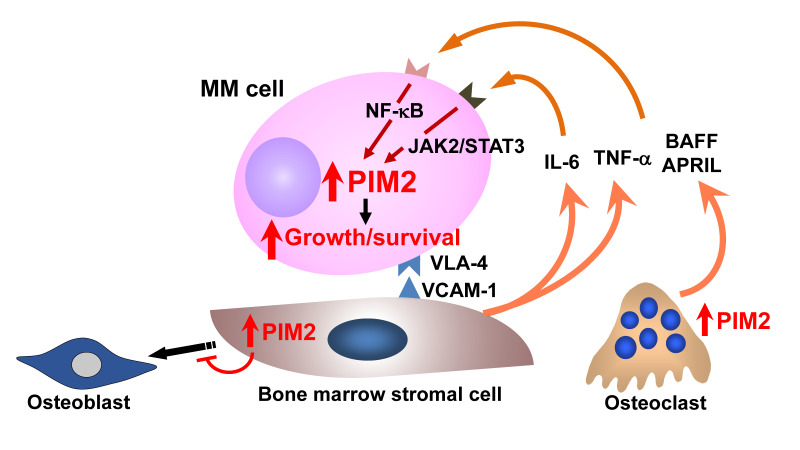
Enhanced expression of PIM2 in MM bone lesions. PIM2 is a novel pro-survival mediator for MM cells. Interaction with MM bone marrow microenvironment potentiates PIM2 expression in MM cells through activation of the JAK2/STAT3 pathway for IL-6 and the NF-κB pathway for TNF family cytokines, TNFα, BAFF, and APRIL, to promote MM cell growth and survival. At the same time, PIM2 is induced in osteoclasts and bone marrow stromal cells through the interaction with MM cells to cause bone destruction. Therefore, PIM2 is overexpressed in MM cells, and bone marrow microenvironment in MM appears to be an important therapeutic target.

**Figure 3 cancers-13-04441-f003:**
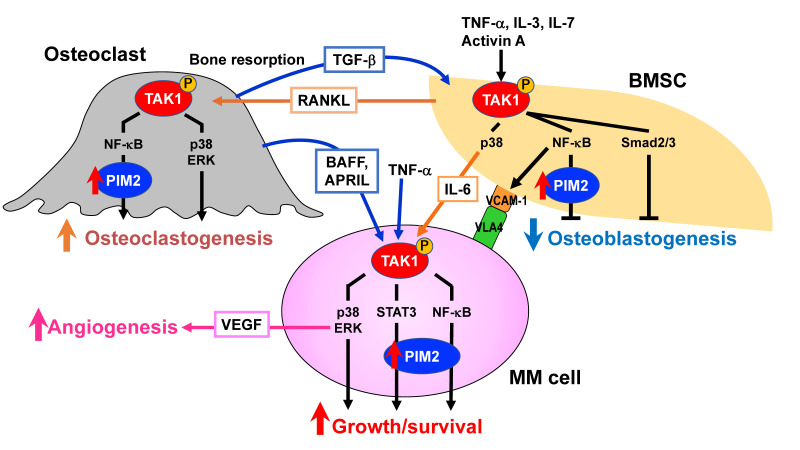
TAK1 as a pivotal therapeutic target in MM. Bone marrow microenvironment is skewed in MM, which confers tumor aggressiveness and progressive bone destruction, posing highly unmet issues. TAK1 mediates a wide range of intracellular signaling pathways. TAK1 is constitutively overexpressed and phosphorylated in MM cells and further upregulated through interaction with bone marrow stromal cells (BMSCs) and osteoclasts. TAK1 mediates major growth and survival signaling pathways in MM cells. TAK1 activation also enhances VEGF production via ERK. TAK1 phosphorylation is also induced to upregulate the expression of vascular cell adhesion molecule-1 (VCAM-1) in BMSCs in the presence of MM cells, which facilitates MM cell-BMSCs adhesion while inducing IL-6 and receptor activator of NF-κB ligand (RANKL) expression by BMSCs. RANKL activates TAK1 in osteoclast precursors to induce osteoclastogenesis. Furthermore, TGF-β released from bone matrices through enhanced bone resorption, activin A produced by environmental cells, and MM cell-derived factors, including IL-3, IL-7, and TNFα, trigger TAK1 phosphorylation in BMSCs to block their osteoblastogenesis. PIM2 is upregulated downstream of TAK1. Therefore, TAK1 activation is vital for MM cell growth and survival and bone destruction. A novel anti-MM treatment with bone modifying actions by TAK1 inhibition can be envisioned.

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
