# Peer review of "Myeloma–Bone Interaction: A Vicious Cycle via TAK1–PIM2 Signaling"

_cancers, 2021, doi:10.3390/cancers13174441_

Round 1
Reviewer 1 Report
The authors have asked all my suggestions
Author Response
The reviewer 1 has raised no more comments and suggestions.
Reviewer 2 Report
Line 406-417: The reference 110 seems wrong. Please check it out
In the section "induction of bone formation", the authors did not mention the bone anabolic effect effects of proteasome inhbitors of osteocytes viability in MM (in vivo and in vitro).
Author Response
Reviewer 2
We thank the reviewer for pointing out our mistakes and making the valuable suggestion.
Line 406-417: The reference 110 seems wrong. Please check it out
Our response:
As the reviewer correctly pointed out, we amended the references #109 and 110 as follows;
- Puchades-Carrasco, L., R. Lecumberri, J. Martínez-López, J. J. Lahuerta, M. V. Mateos, F. Prósper, J. F. San-Miguel and A. Pineda-Lucena. "Multiple myeloma patients have a specific serum metabolomic profile that changes after achieving complete remission." Clin Cancer Res 19 (2013): 4770-9. 10.1158/1078-0432.Ccr-12-2917.
- Bolzoni, M., M. Chiu, F. Accardi, R. Vescovini, I. Airoldi, P. Storti, K. Todoerti, L. Agnelli, G. Missale, R. Andreoli, et al. "Dependence on glutamine uptake and glutamine addiction characterize myeloma cells: A new attractive target." Blood 128 (2016): 667-79. 10.1182/blood-2016-01-690743.
Also, we added these references at appropriate positions in the revised manuscript as follows: “Gln levels have been demonstrated to be decreased in association with increased glutamate levels in bone marrow plasma in patients with active MM, indicating enhanced glutaminolysis [109]. Previous works nicely demonstrated the critical roles of MM-dependent alterations of Gln metabolism in the development of MM bone disease. MM cells appear to be deficient in glutamine synthetase, and highly consume ambient Gln to reduce Gln levels in the bone marrow [110]. Gln also contributes to the energy production necessary for the active biosynthesis of the osteogenic matrix in OB precursors, and Gln deprivation suppresses in vitro OB differentiation of MSCs in osteogenic media. Importantly, OB differentiation is impaired even with slight reduction of Gln content in cultures from 0.6 mM, the physiological level, to 0.4 mM, the average level of MM BM plasma, suggesting a critical involvement of such skewed Gln metabolism in patients with active MM in progression of bone destruction.” (Section 4.2., pages 9-10, lines 400-412)
In the section "induction of bone formation", the authors did not mention the bone anabolic effect effects of proteasome inhbitors of osteocytes viability in MM (in vivo and in vitro).
Our response:
We added the description in the section "induction of bone formation"as follows: “Furthermore, bortezomib has been reported to maintain osteocyte viability and improve bone integrity in bone samples from MM patients [35]. A higher reduction of autophagic osteocytes was observed in MM patients treated with bortezomib-based regimens compared with those treated without bortezomib. Bortezomib was also found to increase the level of LC3II/I ratios and block the degradation of p62 in osteocyte-like cells in cocultures with MM cells as well as upon treatment with high-dose dexamethasone, suggesting suppression of the autophagic death of osteocytes by bortezomib. Therefore, proteasome inhibitors appear to maintain osteocyte viability and thereby improve bone integrity in MM patients.”
(Section 4.3., page 10, lines 445-453).
This manuscript is a resubmission of an earlier submission. The following is a list of the peer review reports and author responses from that submission.
Round 1
Reviewer 1 Report
This is a comprehensive review of the basic science of myeloma bone disease. It is written by authors who work in and understand the field. It serves as a useful review of this important subject.
Author Response
No specific comment is raised.
Reviewer 2 Report
This is an interesting article, the first review for TAK1 and multiple myeloma. Regarding PIM1 there is only another review focused in cancer but not only in MM.
In my opinion, the abstract is too general and do not reflect some aspects reviewed in the manuscript (for example, the signaling pathways described in sections 2-4.2).
Also, figure 1 could be more detailed and include some of signaling pathways mentioned in the text.
In my point of view, drug resistance is a very interesting problem indicated in the manuscript and so, a table summarizing the drug targets mentioned to solve resistance could be very informative.
As the novelty of the manuscript is the role of PIM1 and TAK1, this section could be expanded.
Finally, these works are not cited in the text and could expand the understanding about the role of PIM1 and TAK1 in MM:
TAK1 inhibition subverts the osteoclastogenic action of TRAIL while potentiating its antimyeloma effects.
Tenshin H, Teramachi J, Oda A, Amachi R, Hiasa M, Bat-Erdene A, Watanabe K, Iwasa M, Harada T, Fujii S, Kagawa K, Sogabe K, Nakamura S, Miki H, Kurahashi K, Yoshida S, Aihara K, Endo I, Tanaka E, Matsumoto T, Abe M.Blood Adv. 2017 Oct 26;1(24):2124-2137. doi: 10.1182/bloodadvances.2017008813. eCollection 2017 Nov 14.
Long non-coding RNA LINC01003 suppresses the development of multiple myeloma by targeting miR-33a-5p/PIM1 axis.
Wu L, Xia L, Chen X, Ruan M, Li L, Xia R.Leuk Res. 2021 Mar 31;106:106565. doi: 10.1016/j.leukres.2021.106565. Online ahead of print.PMID: 33865032
Preclinical characterization of INCB053914, a novel pan-PIM kinase inhibitor, alone and in combination with anticancer agents, in models of hematologic malignancies.
Koblish H, Li YL, Shin N, Hall L, Wang Q, Wang K, Covington M, Marando C, Bowman K, Boer J, Burke K, Wynn R, Margulis A, Reuther GW, Lambert QT, Dostalik Roman V, Zhang K, Feng H, Xue CB, Diamond S, Hollis G, Yeleswaram S, Yao W, Huber R, Vaddi K, Scherle P.PLoS One. 2018 Jun 21;13(6):e0199108. doi: 10.1371/journal.pone.0199108. eCollection 2018.PMID: 29927999
Reviewer 3 Report
The review is well written. Since there are a number of reviews on the same topic, the proposed review must be improved.
I suggest the authors to add a paragraph describing the role of myeloma metabolism on osteoblast inhibition. Moreover, some experimental approaches targeting MM metabolism have been proposed.
Additionally, some other aspects of myeloma-bone interaction should be discussed. Among these, the role of miRNA and extracellular vesicles. Moreover, the authors should provide a brief description of the effects of myeloma-bone interaction on immune homeostasis.
The authors also described the role of endothelial cells. The review could be completed by adding the description of adipocytes and their role in MM survival and osteoblast inhibition.
The section of osteocytes is very short. Please improve it.
The authors focused on Pim and Tak. There are many other factors involved in the MM-bone interaction that should be mentioned.Otherwise, the authors should take into consideration to change the title of the review.
I also suggest the suggest the authors to add the main therapeutic options currently available
.
